# Effect of Mealworm Powder Substitution on the Properties of High-Gluten Wheat Dough and Bread Based on Different Baking Methods

**DOI:** 10.3390/foods11244057

**Published:** 2022-12-15

**Authors:** Xinyuan Xie, Ke Cai, Zhihe Yuan, Longchen Shang, Lingli Deng

**Affiliations:** College of Biological and Food Engineering, Hubei Minzu University, Enshi 445000, China

**Keywords:** wheat dough, mealworm powder, pasting, farinograph, extensograph, bread, baking methods

## Abstract

Mealworms (*Tenebrio molitor*) are protein-rich edible insects that have been regarded as novel food ingredients. In this study, high-gluten wheat flour was formulated with dried mealworm powder at various levels (0%, 5%, 10%, 15%, and 20%) to study its influence on the pasting, farinograph, and extensograph properties and microstructure of the dough. A subsequent decrease in the pasting parameters was observed due to starch dilution. The water absorption, dough development time, and dough stability time decreased gradually from 71.9% to 68.67%, 13.6 min to 10.43 min, and 14.1 min to 5.33 min, respectively, with the increase in the substitution of mealworm powder from 0% to 20%. The farinograph characteristics corresponded to a weak gluten network formed through the dilution of gluten by the replacement of wheat flour with a non-gluten ingredient. The stretch ratio of the high-gluten dough increased gradually from 4.37 (M0) to 6.33 (M15). The increased stretching resistance and extensibility of the dough with 5% and 10% mealworm powder indicated that mealworm powder can act as a plasticizer in the gluten network, which might contribute to the decreased strength and increased elasticity and flexibility of the dough network. The bread made with three different baking methods showed similar increases in specific volume and decreased hardness up to the 10% substitution level, owing to the increased elasticity and flexibility of the dough. The GB/T 35869-2018 Rapid-baking method, GB/T 14611-2008 Straight dough method, and automatic bread maker method exhibited the highest specific volumes of 3.70 mL/g, 3.79 mL/g, and 4.14 mL/g when the wheat flour was substituted with 10% mealworm powder. However, 15% and 20% mealworm powder substitution markedly reduced the bread quality owing to the dilution effect and mealworm powder phase separation. These results provide a perspective on the relationship between the rheological properties of mealworm powder-substituted high-gluten dough and application suggestions for insect food development in the food industry.

## 1. Introduction

To deal with the global population increase, the loss of farmland, and global climate change, there is an urgent need for alternative protein sources to replace animal protein production [1]. Proteins of vegetable origin were identified as the first candidates, but they are not the best choices owing to the low protein and limited amino acid contents of plants [2]. Insects have traditionally been regarded as a part of the diet in most countries. In comparison to the proteins of plant and common livestock origin, those from insects have the advantages of being environmentally friendly, requiring less land and feed use, and having a high food conversion ratio. From a nutritional perspective, insects have a high content of protein (35–61%), lipids (13–33%), and dietary fiber. In addition to macronutrients, some insects are also known for their rich mineral and vitamin profiles [3]. The yellow mealworm (*Tenebrio molitor*), as a commonly farmed insect used for pet food, has attracted attention in the food industry. The dried yellow mealworm was regarded as a novel food by the European Food Safety Authority in 2021, according to Regulation (EU) 2015/2283 [4]. Mealworm larva powder can provide not only up to 50% protein but also up to 28% lipids, which include essential fatty acids and essential amino acids [2,5]. Some previous studies and commercial attempts have been conducted to apply mealworm powder to bakery foods to enrich their nutrition and improve the sensory properties of such products, including bread [5,6], cookies [7], biscuits [8], pasta [9], and chips [10].

Among flour-based foods, bread is the most widely consumed bakery food in terms of daily diets [11,12,13,14]. Some previous studies have focused on nutritional improvements to bread using mealworm flour [15]. Roncolini et al. [6] supplemented soft wheat flour with 5% and 10% mealworm (*Tenebrio molitor*) powder, and the bread fortified with mealworm powder exhibited a significant increase in protein content, essential amino acids, and essential fatty acids [2]. Kowalski et al. [5] added cricket, buffalo worm, and mealworm flour to bread and found that the amino acid score of lysine increased from 40% to nearly 70% with 10% insect powder, as compared to the score in traditional wheat bread. Further, Roncolini et al. [6] showed that 10% mealworm powder could result in a significant increase in the contents of tyrosine, methionine, isoleucine, and leucine. Moreover, González et al. [16] revealed the effect of three types of insect flour on bread quality and found that wheat flour replacement (5%) with insect flours from *Hermetia illucens*, *Acheta domestica*, and *Tenebrio molitor* improved the nutritional value of bread, especially the protein content. Gaglio et al. [17] suggested mealworm powder enrichment reduced the starch digestibility of the sourdough “ciabatta” bread, indicating its potential as a low glycemic food.

Even though mealworm powder-supplemented bread has been studied by some researchers, the interaction between mealworm powder and the dough network is still unclear. Obviously, the nutritional properties can be enhanced through mealworm powder addition, but the effect of mealworm powder on the processing properties of bread should be considered. Mealworm powder could interact with gluten and the dietary fiber, proteins, and lipids, facilitating or disturbing the formation of the gluten network and gas retention ability, and resulting in a change in the bread quality [18]. In previous studies, the effect of mealworm powder on the bread quality varied, which might be ascribed to the various production methods and manual factors during bread making. An automatic bread maker is a common bread-making appliance, which can be applied to bread-making in the laboratory to eliminate manual factors [19]. Hence, to verify the feasibility of the automatic bread-making method in a bakery study, the commonly used GB/T 14611-2008 Straight dough method [20] and the newly established GB/T 35869-2018 Rapid-baking test method [21] were chosen as references.

In this study, we hypothesized that mealworm substitution would affect the pasting characteristics and the farinograph and extensograph properties of the wheat dough, resulting in a bread quality change. To further explore the effect of mealworm substitution and baking methods on the physical properties of bread, it was prepared following the GB/T 35869-2018 Rapid-baking test method, GB/T 14611-2008 Straight dough method, and automatic bread maker method. Accordingly, specific volume, porosity, colorimetric, and texture analyses of the bread were conducted.

## 2. Materials and Methods

### 2.1. Materials

The high-gluten wheat flour (protein content of 12.8%) was supplied by Xinxiang Xinliang Cereals Processing Co., Ltd. (Xinxiang, China). Mealworm powder was supplied by Qingdao Sino Crown Biological Engineering Co., Ltd. (Qingdao, China). Butter and yeast were purchased from Angel Yeast Co., Ltd. (Wuhan, China). Salt and sugar were obtained from a local supermarket.

### 2.2. Pasting Characteristics

The mealworm was incorporated into high-gluten wheat flour at 0% (M0), 5% (M5), 10% (M10), 15% (M15), and 20% (M20) concentrations. Per the American Association of Cereal Chemists (AACC) method 76–21 (AACC, 2000), a Rapid Visco Analyzer (RVA) (RVA-Eritm, Perkin Elmer, Waltham, MA, USA) was used to study the pasting properties of the control wheat flour and the blends [22].

### 2.3. Farinographic and Extensograph Properties

The mixing properties of control wheat flour and blends were analyzed according to the AACC Method 54-21 (AACC, 2000). The flour sample was tested using a farinograph (JFZD, Beijing Dongfu Jiuheng Instrument Technology Co., Ltd., Beijing, China) [23]. Dough elastic properties were tested using a JMLD150 Extensograph (Dongfu, Beijing, China).

### 2.4. Scanning Electron Microscopy (SEM)

The wheat dough prepared from the mixed flour was dried at −80 °C overnight, and the samples were then sputtered with a thin layer of gold under vacuum conditions. The freeze-dried dough was observed using a scanning electron microscope (TESCAN MIRA LMS, Brno–Kohoutovice, Czech Republic) at an acceleration voltage of 15 kV.

### 2.5. Bread Preparation

The bread was prepared following the GB/T 35869-2018 Rapid-baking test method, in accordance with the GB/T 14611-2008 Straight dough method, and with an automatic bread maker (MM-ESC1510, Midea, Foshan, China). Formulations are shown in detail in Table 1.

GB/T 35869-2018 Rapid-baking method [21]: All raw materials were mixed and kneaded for 20 min using a dough mixer (AM-CG108-1, ACA). The resultant dough was then subjected to pre-fermentation at 38 °C with 85% relative humidity for 20 min in a controlled fermentation cabinet (DHTHM-16-0-P-SD, Doaho Test Co., Ltd., Shanghai, China). The fermented dough was divided into several pieces with a weight of approximately 75 g, and each piece was then molded into a round shape. After fermentation (40 min, 38 °C, 85% relative humidity), the bread was baked in a steam oven (K6, Daewoo) at 175 °C for 20 min.

GB/T 14611-2008 Straight dough method [20,24]: All raw materials were mixed thoroughly using a dough mixer (AM-CG108-1, ACA) for 20 min. The dough was leavened for 70 min at 30 °C in a glass bowl sealed with food-grade cling wrap (DHTHM-16-0-P-SD, Doaho Test Co., Ltd., Shanghai, China). Thereafter, the fermented dough was divided into pieces weighing approximately 75 g, and each piece was formed into a round shape and sealed with food-grade cling wrap for secondary fermentation (20 min at 30 °C). The bread was baked at 175 °C with top and bottom heat for 20 min in a steam oven (K6, Daewoo).

Automatic bread maker method [25]: To prepare the bread, 250 g of mixed flour, 15 g of butter, 90 g of water, 18 g of sugar, 3 g of salt, 12 g of milk powder, and 3 g of yeast were put into a bread maker (MM-ESC1510, Midea). The wheat flour was replaced at ratios of 0%, 5%, 10%, 15%, and 20% with mealworm powder to make the bread. The operating conditions of the bread maker were set as basic bread, light color, and 500 g.

### 2.6. Physiochemical Analyses of Bread

#### 2.6.1. Specific Volume, Porosity, and Colorimetric Analyses

The specific volume of bread was determined using the millet displacement method. The specific volume (mL/g) was calculated as the volume (mL)/weight (g) of bread. Image analysis was employed for the evaluation of porosity and cell density (crumb porosity), according to the method of Kowalski et al. [5]. Images of the bread cross-sections were obtained, and the porosity of the bread was analyzed using ImageJ software. The images were converted into 8-bit gray images, and the corresponding regions were selected for analysis to obtain the cross-sectional porosity.

The crumb colors of the bread samples were detected using a colorimeter (CS-820N, Hangzhou CHNSpec Technology Co., Ltd., Hangzhou, China). The parameters *L** (darkness/brightness), *a** (shade of red/green), and *b** (shade of blue/yellow) were analyzed. The determination was based on the *L**, *a**, and *b** color systems.

#### 2.6.2. Texture

Texture profile analysis (TPA) was performed using a TA-XT plus texture analyzer (Stable Micro Systems Ltd., Surrey, UK) with a 36 mm diameter cylinder probe P/36R, according to the method described by de Castro et al. [26]. The bread was cut into pieces with a thickness of 5 cm. The pre-test, test, and post-test speeds were 2 mm/s, and the sample compression was 25% of its original height with a 5.0 g trigger force.

### 2.7. Statistical Analysis

All experiments were performed at least in triplicate. OriginPro 2022b (OriginLab, Northampton, MA, USA) was used for data processing. Data were expressed as the mean ± standard deviation. Statistical differences in the correlation analysis were determined by performing a one-way analysis of variance (ANOVA). Tukey’s test at *p* < 0.05 was used to determine significant differences between mean values.

## 3. Results and Discussion

### 3.1. Pasting Properties

The RVA curves and pasting parameters are shown in Figure 1 and Table 2, respectively. The pasting parameters are related to the extent of starch granule swelling [27]. The peak viscosity of the high-gluten wheat flour decreased nearly linearly with an increase in the mealworm powder content, from 1073.67 cp for wheat flour (M0) to 580.33 cp for that formulated with 20% mealworm powder (M20). The other parameters showed similar decreasing trends, which were ascribed to the starch dilution effect mediated by mealworm powder substitution [28]. Khuenpet et al. [29] also observed similar decreasing trends in peak viscosity and breakdown viscosity. The increased pasting temperature of the wheat flour with mealworm powder might be attributed to the slow heat transfer of the mixed flour due to the increase in fat and dietary contents [30].

### 3.2. Farinograph Properties

The farinograph curves and parameters are shown in Figure 2 and Table 3, respectively. Water absorption is a parameter that is indicative of the ability of the mixed flour to absorb water and form dough of optimal consistency. The water absorption decreased from 71.9% for the control (M0) to 68.67% for the wheat flour formulated with a 20% mealworm substitution (M20). This decrease in water absorption was ascribed to the gluten-dilution effect, which is consistent with the pasting results. A previous study has reported that the nutritional components of mealworm powder include 4.43% water, 3.5% ash, 1.17% carbohydrate, 43.5% protein, 25.3% fat, and 22.1% dietary fiber [8]. It can be inferred that the high content of fat and dietary fiber retarded the water absorption of the mixed flours [31]. Waseem et al. [32] also observed a similar decrease in water absorption when wheat flour was substituted with 20% spinach powder. Fang et al. [33] also attributed a 10% isomaltodextrin substitution resulting in decreased water absorption to the incorporation of high dietary fiber content.

The dough development time, stability time, and farinograph quality number are usually positively related to the dough strength [23]. The dough development time decreased significantly from 13.6 min for M0 to approximately 11 min for the dough with various mealworm powder contents. Furthermore, the stability time showed a substitution level-dependent decrease. This decreased tolerance to the mixing of high-gluten dough can be partially explained by the interactions between mealworm powder and gluten, which prevent the complete hydration of proteins, thereby impeding the proper formation of the high-gluten matrix during dough mixing. The mealworm powder might act as a plasticizer of the gluten network, potentially weakening the interactions between glutenin chains. Similar tendencies, such as a decrease in dough stability following supplementation with dietary fiber-rich plant materials, have also been reported by other researchers. Tian et al. [27] observed an increasing trend in the dough development time and stability time upon adding a certain amount of *Chlorella pyrenoidosa* powder to high-gluten wheat flour (11.76 g/100 g protein). Further, Zarzycki et al. [34] found that the dough stability of high-gluten wheat flour decreased from 12.7 min (control) to 10.5 min for wheat flour formulated with 12 g/100 g defatted Moldavian dragonhead seed residue flour. The farinograph quality number, which is a measure of the ability of the dough to retain its structure over time during mixing, showed a change similar to that observed in the stability time values. A low degree of softening is desirable and indicates a good tolerance of the dough to mixing. The degree of softening of high-gluten dough increased significantly after mealworm substitution, which corresponded to the weak gluten network resulting from the dilution of gluten by replacing wheat flour with non-gluten ingredients. Similar to the present study, a previous study reported an increase in the degree of high-gluten dough softening after dough supplementation with 0.5% *Chlorella pyrenoidosa* powder [27]. The color of the dough was influenced by the addition of mealworm powder. Owing to the brown color of the mealworm, the lightness was significantly decreased after this substitution. Moreover, the redness (*a**) increased significantly, and the yellowness (*b**) showed an increasing trend.

### 3.3. Extensograph and Texture Properties

The farinograph properties reflect the strength of the wheat dough, whereas the extensograph properties reflect the viscoelastic properties of the dough [35]. The extensograph curves are presented in Figure 3, with the extensograph parameters shown in Table 4. The total energy required to stretch the dough from the beginning of stretching to breaking is expressed as the stretching energy. The dough mixed with a 5% mealworm powder substitution gradient showed significantly increased stretching energy and stretching resistance compared to the dough without mealworm powder. This result could be attributed to the interactions between mealworm proteins, lipids, and dietary fiber and gluten proteins. With an increasing substitution gradient, the stretching resistance showed an increasing trend up to a 15% substitution; however, no significant difference was observed among the dough samples with different substitution gradients. The extensibility of the high-gluten dough formulated with or without mealworm powder showed no significant changes between samples. Zhang et al. [36] found that bamboo shoot dietary fiber increased the tensile resistance and improved the viscoelasticity, extensibility, and plasticity of wheat dough. The rearrangement of starch granules and mealworm powder within the dough structure during extensional deformation could interfere with the protein network [37]. The high-gluten dough showed decreased strength and resistance to mixing as well as high stretching resistance and extensibility. Based on these findings and observations, it could be inferred that mealworm powder can act as a plasticizer in the gluten network, which might contribute to the diminished strength but increased elasticity and flexibility of the dough network.

### 3.4. Dough Microstructure

SEM observations were used to investigate the continuity of the wheat dough with various levels of mealworm powder (Figure 4). The starch granules were embedded in the gluten network of the wheat dough without mealworm powder (M0). Continuous gluten networks containing starch granules and gluten films were observed in mealworm-free and 5% and 10% mealworm-substituted dough samples, probably because the small amount of mealworm powder components could properly fill the gap between the gluten network and act as a plasticizer. Upon substitution with up to 15% and 20% mealworm powder, some starch granules and mealworm powder particles were released from the gluten network, causing it to become discontinuous. Accordingly, it has been reported that insoluble fiber can destroy the network structure of gluten [38].

### 3.5. Physical Properties of Bread

The visual appearances of the bread and crumb and the binarized images of the cross-section are shown in Figure 5. The visual appearance gave the initial impression that the bread made using the high-gluten wheat flour formulated with 20% mealworm powder (M20) was unacceptable to consumers owing to the dark color and decreased volume. Table 4 shows the *L** (lightness), *a** (redness), *b** (yellowness), specific volume, and porosity values calculated from the binarized images. As the mealworm powder substitution level was increased, the *L** value of bread decreased, whereas the redness and yellowness increased accordingly. The color change could be ascribed to the incorporation of brown mealworm powder and the occurrence of the Maillard reaction.

As can be seen from Table 5, with an increase in the mealworm powder content, the specific volume of bread increased by up to 10% and then decreased gradually. The bread produced using all three methods showed the highest specific volume with a 10% mealworm powder substitution. The GB/T 35869-2018 Rapid-baking method, GB/T 14611-2008 Straight dough method, and automatic bread maker method exhibited the highest specific volumes of 3.70, 3.79, and 4.14 mL/g, respectively. Even though a subsequent decrease in the pasting parameters was observed due to starch dilution. The farinograph properties corresponded to a weak gluten network formed through the dilution of gluten by the replacement of wheat flour with a non-gluten ingredient. The high-gluten dough showed increased stretching resistance and extensibility. It can be inferred that mealworm powder can act as a plasticizer in the gluten network, which might contribute to the decreased strength and increased elasticity and flexibility of the dough network, resulting in an increased gas retention ability at 5% and 10% mealworm substitution levels. When the substitution level exceeded 10%, the specific volume decreased, indicating that the high level of mealworm substitution decreased the bread dough strength and gas retention stability. At substitution levels of up to 20%, the continuity of the bread dough might have been affected due to the sedimentation of mealworm powder. The porosity of the bread based on cross-sections showed a trend similar to that of the specific volume. The specific volume and porosity commonly exhibited positive correlations with the bread quality. However, compared to the bread with a 15% substitution level, the bread with a 20% substitution level showed an increasing trend in porosity with all production methods.

Roncolini et al. [6] also observed a similar increase in the specific volume after substituting soft wheat flour with 5% and 10% mealworm powder; they found that the 5% substitution level achieved the highest specific volume, but they did not study a substitution level higher than 10%. It is known that mealworm powder is rich in not only protein but also lipids, which can be adsorbed at the gas cell–dough interface and then increase gas retention during leavening [39]. However, some previous studies have shown the opposite effect of mealworm powder on the bread-specific volume. Khuenpet et al. [29] found that the specific volume of bread decreased gradually with an increase in mealworm substitution of up to 15% and that the hardness of the bread increased by approximately four times compared with the control bread. González et al. [16] used mealworm powder to replace 5% of the wheat flour and observed a significantly decreased specific volume compared to the control. However, by examining the bread-making method used in previous research, it can be inferred that the opposite trend was attributed to an inappropriate dough mixing time; Khuenpet et al. [29] mixed ingredients using a food mixer machine for 30 min, whereas González et al. [16] mixed the dough for only 5 min. Another noteworthy point is whether adjusting the water content in bread making is suggestive of the farinograph property. In the present study, the water content was adjusted according to the farinograph property for the GB/T 35869-2018 Rapid-baking method and GB/T 14611-2008 Straight dough method, whereas the water content remained the same for the automatic bread maker method. The similar trend in the specific volume and porosity indicates that the change in bread quality is less susceptible to an adjustment of the water content.

### 3.6. Texture Analysis

Texture parameters of the bread were shown in Table 6. Hardness is generally negatively related to the bread quality. With the three different baking methods, the hardness of the bread decreased significantly with up to 10% mealworm substitution and then increased significantly for the bread with a 15% mealworm substitution. For the straight dough method, the hardness of the bread decreased significantly from 415 g (M0) to 350 g (M5). The bread made by the automatic bread maker showed relatively low hardness compared to the other methods, which might be due to the increased portion of butter and sugar in the ingredients. The hardness significantly decreased to 262 g (M5) and 179 g (M10) compared to the control M0 (324 g), which become coccoid with the highest specific volume (4.41 mL/g) of the bread made by the bread maker with 10% mealworm powder. Interestingly, the hardness of the bread with 20% mealworm powder was decreased slightly compared to that with 15%, which might be ascribed to the phase separation of mealworm powder and bread dough. Roncolini et al. [6] also found that the addition of 5% and 10% mealworm powder yielded softer bread, as indicated by the significantly decreased hardness compared with the control bread. However, González et al. [16] observed no significant differences in the texture parameters between the bread with 5% mealworm powder and the control. Higher specific volume is usually related to the decreased hardness of bread. Khuenpet et al. [29] found that the hardness of bread increased gradually with an increase in mealworm substitution of up to 15%. The chewiness and gumminess of the bread with various methods showed a similar trend to that of hardness. The cohesiveness of the bread showed no significant change with up to a 15% mealworm substitution level, indicative of the continuity of the bread dough network, which was coccoid based on SEM observations. The cohesiveness of bread with 20% mealworm powder using the GB/T 35869-2018 method was decreased significantly to 0.564. The resilience showed no significant change with up to a 5% mealworm substitution for the GB/T 35869-2018 Rapid-baking method and GB/T 14611-2008 Straight dough method, whereas with the automatic bread maker method, the resilience was maintained at up to a 10% substitution level. Compared with the other two standard bread-making methods, the automatic bread maker results showed a similar trend in the physical properties and better bread quality, indicating its potential as an alternative method in bakery studies.

## 4. Conclusions

Given the present results, it was found that the mealworm substitution (0–20%) had a negative impact on the pasting and farinograph characteristics, owing to the starch and gluten dilution effects. At low substitution levels (5% and 10%), mealworm powder showed a positive impact on the extensograph characteristics and bread quality (increased specific volume and decreased hardness). However, further higher substitution levels (15% and 20%) deteriorated the elasticity and bread quality (decreased specific volume and increased hardness). The microstructure of the gluten network was not significantly affected by up to 15% mealworm substitution. Overall, a 10% mealworm substitution level seems to be a suitable choice to improve the physical and nutritional properties of bread. The GB/T 35869-2018 Rapid-baking test method, GB/T 14611-2008 Straight dough method, and automatic bread maker method showed similar trends in this study. Thus, the automatic bread maker method might be an effective method for bakery studies to avoid manual variance. These results provide a perspective for the application of insects as protein alternatives and the development of insect-based foods in the food industry.

## Figures and Tables

**Figure 1 foods-11-04057-f001:**
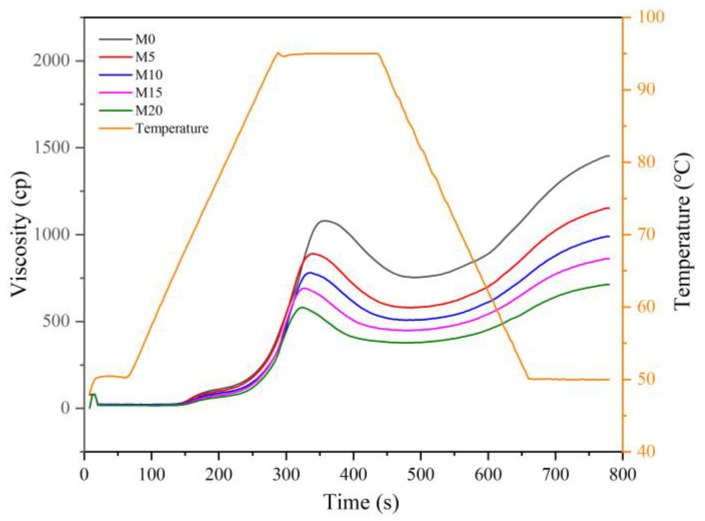
Rapid Visco Analyzer (RVA) curves of high-gluten wheat flour formulated with 0% (M0), 5% (M5), 10% (M10), 15% (M15), and 20% (M20) mealworm powder.

**Figure 2 foods-11-04057-f002:**
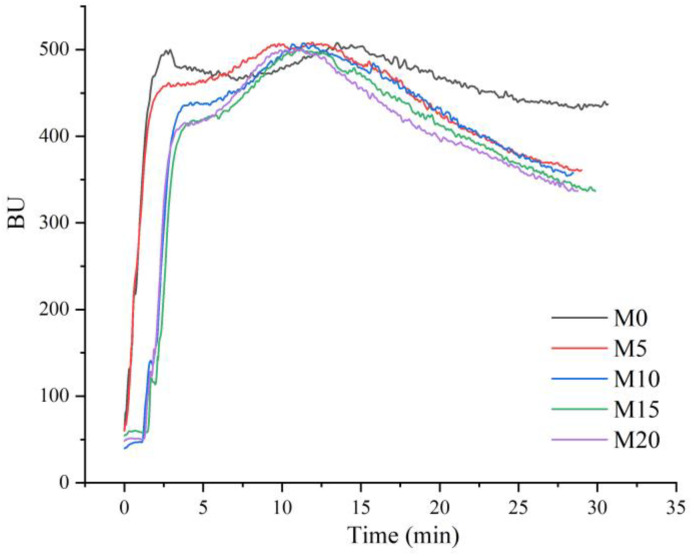
Farinograph curves of high-gluten wheat flour formulated with 0% (M0), 5% (M5), 10% (M10), 15% (M15), and 20% (M20) mealworm powder. BU: Brabender unit.

**Figure 3 foods-11-04057-f003:**
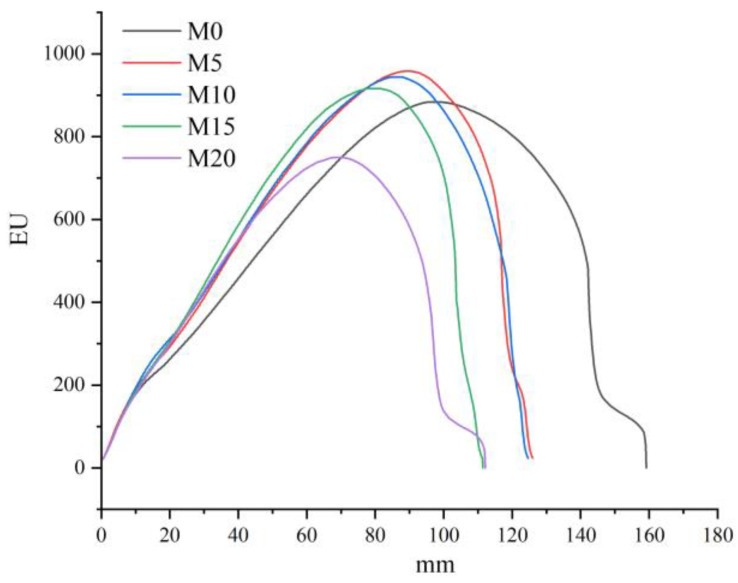
Extensograph curves of high-gluten wheat dough formulated with 0% (M0), 5% (M5), 10% (M10), 15% (M15), and 20% (M20) mealworm powder with a resting time of 45 min. EU: extensograph units.

**Figure 4 foods-11-04057-f004:**
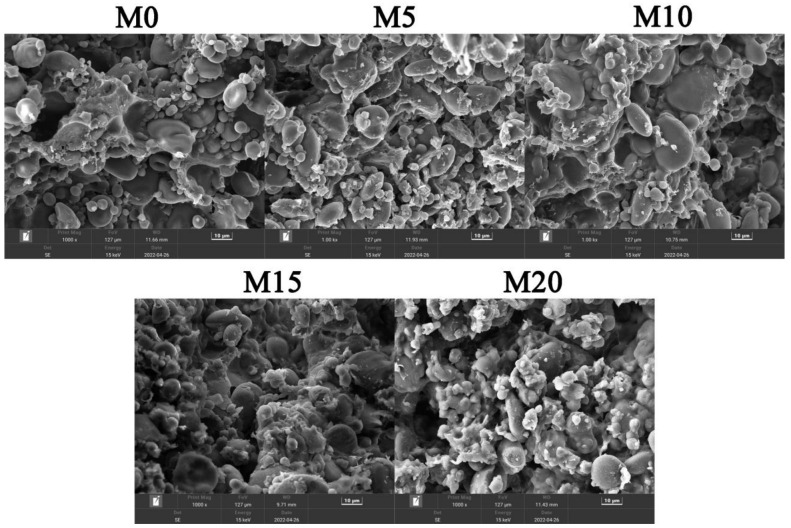
Representative scanning electron microscopy images of high-gluten wheat flour with 0% (M0), 5% (M5), 10% (M10), 15% (M15), and 20% (M20) mealworm powder substitution levels.

**Figure 5 foods-11-04057-f005:**
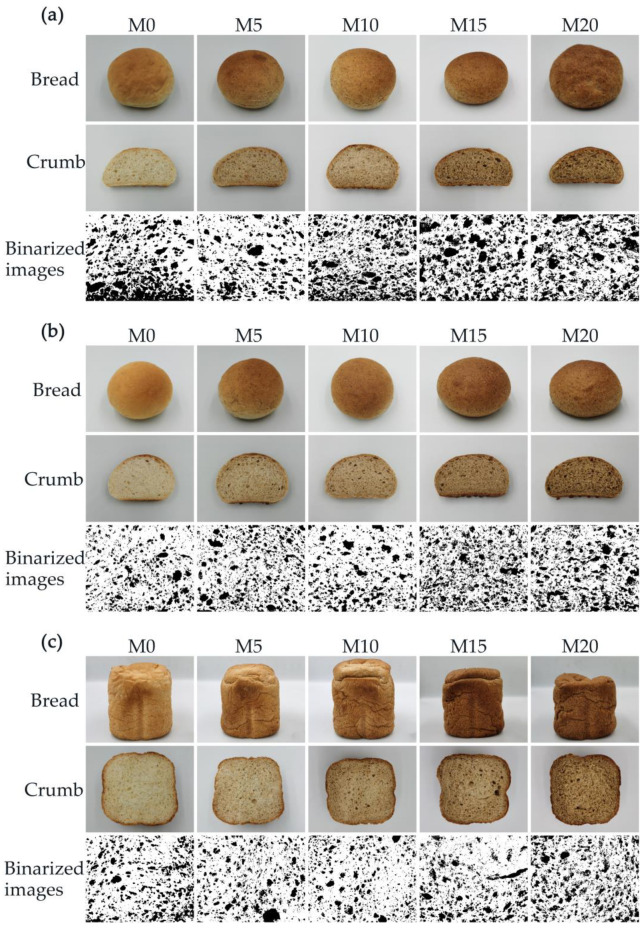
Visual appearance of the bread and crumb and the binarized images of the cross-sections with 0% (M0), 5% (M5), 10% (M10), 15% (M15), and 20% (M20) mealworm powder: (**a**) GB/T 35869-2018 Rapid-baking method; (**b**) GB/T 14611-2008 Straight dough method; (**c**) automatic bread maker method.

**Table 1 foods-11-04057-t001:** Formulas for bread-making methods with added mealworm powder.

Bread-Making Method	Sample	Flour (g)	Mealworm Powder (g)	Yeast (g)	Salt (g)	Sugar (g)	Milk Powder (g)	Butter (g)	Water ^a^ (mL)
GB/T 35869-2018	M0	100	-	2.7	1.5	6	-	3.0	70
	M5	95	5	2.7	1.5	6	-	3.0	69
	M10	90	10	2.7	1.5	6	-	3.0	68
	M15	85	15	2.7	1.5	6	-	3.0	67.5
	M20	80	20	2.7	1.5	6	-	3.0	66.6
GB/T 14611-2008	M0	100	-	1.8	1.5	6.0	4.0	3.0	70
	M5	95	5	1.8	1.5	6.0	4.0	3.0	69
	M10	90	10	1.8	1.5	6.0	4.0	3.0	68
	M15	85	15	1.8	1.5	6.0	4.0	3.0	67.5
	M20	80	20	1.8	1.5	6.0	4.0	3.0	66.6
Bread machine	M0	250	-	3	3	18	12	16	170
	M5	237.5	12.5	3	3	18	12	16	170
	M10	225	25	3	3	18	12	16	170
	M15	212.5	37.5	3	3	18	12	16	170
	M20	200	50	3	3	18	12	16	170

**^a^** The added quantities of water according to the water absorption percentages recorded in farinograph trials. The flour was substituted by mealworm powder at weight ratios of 0% (M0), 5% (M5), 10% (M10), 15% (M15), and 20% (M20), respectively.

**Table 2 foods-11-04057-t002:** Pasting characteristics of high-gluten wheat flour with 0% (M0), 5% (M5), 10% (M10), 15% (M15), and 20% (M20) mealworm powder substitution levels.

	Peak Viscosity (cp)	Trough Viscosity (cp)	Breakdown Viscosity (cp)	Final Viscosity (cp)	Setback (cp)	Peak Time (min)	Pasting Temperature (°C)
M0	1073.67 ± 8.39 ^a^	730.33 ± 29.7 ^a^	343.33 ± 21.5 ^a^	1423.67 ± 44.81 ^a^	693.33 ± 16.5 ^a^	5.89 ± 0.03 ^a^	89.92 ± 0.03 ^a^
M5	887 ± 11.53 ^b^	589.33 ± 17.93 ^b^	297.67 ± 10.02 ^b^	1154.67 ± 27.10 ^b^	565.33 ± 12.42 ^b^	5.76 ± 0.08 ^a^	90.68 ± 0.03 ^a^
M10	782.67 ± 10.6 ^c^	509.33 ± 9.07 ^c^	273.33 ± 1.53 ^b^	992.33 ± 16.62 ^c^	483 ± 7.55 ^c^	5.55 ± 0.04 ^b^	91.23 ± 0.92 ^a^
M15	669 ± 17.09 ^d^	425.67 ± 20.82 ^d^	243.33 ± 5.03 ^c^	822.67 ± 35.47 ^d^	397 ± 15 ^d^	5.42 ± 0.08 ^bc^	91.20 ± 0.91 ^a^
M20	580.33 ± 5.03 ^e^	377 ± 1.73 ^d^	203.33 ± 3.51 ^d^	715.33 ± 4.04 ^e^	338.33 ± 3.51 ^e^	5.35 ± 0.04 ^c^	91.22 ± 0.98 ^a^

Values in the same column followed by different superscript letters are significantly different (*p* < 0.05).

**Table 3 foods-11-04057-t003:** Farinograph properties and color of high-gluten wheat flour with 0% (M0), 5% (M5), 10% (M10), 15% (M15), and 20% (M20) mealworm powder substitution levels.

	Water Absorption (%)	Dough Development Time (min)	Stability Time (min)	Degree of Softening (FU)	Farinograph Quality Number	*L**	*a**	*b**
M0	71.90 ± 0.08 ^a^	13.60 ± 0.08 ^a^	14.10 ± 1.84 ^a^	65.67 ± 6.34 ^b^	175.67 ± 4.99 ^a^	88.09 ± 0.86 ^a^	0.84 ± 0.11 ^d^	15.03 ± 0.42 ^b^
M5	70.60 ± 0.29 ^b^	10.83 ± 0.79 ^b^	8.87 ± 0.05 ^b^	115.67 ± 4.78 ^a^	151.33 ± 5.73 ^b^	81.58 ± 0.49 ^b^	2.00 ± 0.22 ^c^	15.17 ± 0.84 ^b^
M10	70.00 ± 0.08 ^c^	11.20 ± 0.43 ^b^	6.17 ± 0.33 ^bc^	121.67 ± 6.94 ^a^	146.33 ± 4.92 ^b^	73.01 ± 2.72 ^c^	4.16 ± 0.65 ^b^	17.72 ± 1.45 ^a^
M15	69.43 ± 0.09 ^d^	11.03 ± 0.24 ^b^	5.67 ± 0.25 ^c^	118.33 ± 4.03 ^a^	145.33 ± 4.03 ^b^	67.19 ± 0.87 ^d^	5.18 ± 0.27 ^a^	18.68 ± 0.66 ^a^
M20	68.67 ± 0.05 ^e^	10.43 ± 0.62 ^b^	5.33 ± 0.05 ^c^	119.00 ± 4.55 ^a^	137.67 ± 2.05 ^b^	65.32 ± 1.67 ^d^	5.54 ± 0.49 ^a^	19.43 ± 1.35 ^a^

Values in the same column followed by different superscript letters are significantly different (*p* < 0.05). *L**, lightness; *a**, redness; *b**, yellowness.

**Table 4 foods-11-04057-t004:** Extensograph and texture properties of high-gluten wheat flour with 0% (M0), 5% (M5), 10% (M10), 15% (M15), and 20% (M20) mealworm powder substitution levels.

	Stretching Energy (cm^2^)	Extensibility (mm)	Stretching Resistance (BU)	Stretch Ratio	Hardness (g)	Stickiness (g)
M0	120.67 ± 8.34 ^b^	127.67 ± 6.65 ^a^	557.00 ± 50.68 ^b^	4.37 ± 0.52 ^b^	42.742 ± 3.293 ^b^	30.106 ± 3.030 ^bc^
M5	143.00 ± 4.55 ^a^	132.67 ± 11.73 ^a^	696.00 ± 22.05 ^a^	5.30 ± 0.49 ^ab^	53.751 ± 1.226 ^a^	19.949 ± 2.751 ^a^
M10	128.67 ± 5.73 ^ab^	122.67 ± 7.13 ^a^	671.33 ± 25.75 ^a^	5.50 ± 0.54 ^ab^	54.686 ± 2.630 ^a^	30.833 ± 2.872 ^bc^
M15	116.67 ± 6.13 ^b^	111.00 ± 0.82 ^a^	702.33 ± 34.08 ^a^	6.33 ± 0.25 ^a^	57.528 ± 2.560 ^a^	34.879 ± 2.996 ^c^
M20	96.67 ± 4.64 ^c^	114.33 ± 4.99 ^a^	650.00 ± 17.68 ^ab^	5.70 ± 0.14 ^ab^	54.794 ± 2.570 ^a^	24.814 ± 2.146 ^ab^

Values in the same column followed by different superscript letters are significantly different (*p* < 0.05).

**Table 5 foods-11-04057-t005:** The colorimetric properties, specific volumes, and porosity of bread with various mealworm powder substitution levels.

	*L**	*a**	*b**	Specific Volume (mL/g)	Porosity (%)
GB/T 35869-2018					
M0	69.10 ± 1.82 ^b^	−1.04 ± 0.64 ^g^	5.58 ± 2.66 ^i^	3.15 ± 0.15 ^bcde^	36.05 ± 0.57 ^bc^
M5	63.10 ± 1.92 ^c^	−0.16 ± 0.39 ^fg^	10.26 ± 1.47 ^fgh^	3.54 ± 0.25 ^abcd^	23.59 ± 0.32 ^f^
M10	56.71 ± 1.91 ^d^	1.76 ± 1.16 ^cde^	13.35 ± 1.27 ^def^	3.70 ± 0.33 ^ab^	41.29 ± 0.62 ^a^
M15	52.83 ± 2.94 ^def^	2.75 ± 0.11 ^bcd^	16.98 ± 0.45 ^bc^	3.59 ± 0.17 ^abcd^	33.28 ± 0.50 ^d^
M20	49.64 ± 1.92 ^f^	3.84 ± 0.94 ^b^	21.34 ± 1.13 ^a^	3.05 ± 0.33 ^cdef^	41.77 ± 0.85 ^a^
GB/T 14611-2008					
M0	75.03 ± 1.52 ^a^	−1.12 ± 0.13 ^g^	8.31 ± 0.49 ^hi^	3.22 ± 0.11 ^bcdef^	22.27 ± 0.92 ^f^
M5	67.61 ± 1.10 ^b^	0.76 ± 0.32 ^ef^	12.78 ± 1.03 ^defg^	3.47 ± 0.28 ^abcde^	29.36 ± 0.34 ^e^
M10	63.19 ± 0.77 ^c^	2.79 ± 0.93 ^bcd^	15.85 ± 2.68 ^bcd^	3.79 ± 0.17 ^ab^	41.50 ± 0.65 ^a^
M15	56.62 ± 2.14 ^de^	3.64 ± 0.64 ^b^	18.29 ± 1.46 ^ab^	2.43 ± 0.01 ^f^	34.13 ± 2.43 ^cd^
M20	51.88 ± 1.45 ^f^	5.64 ± 0.55 ^a^	21.30 ± 1.71 ^a^	2.74 ± 0.14 ^ef^	37.44 ± 0.46 ^b^
Bread maker					
M0	52.46 ± 2.35 ^ef^	−0.37 ± 0.69 ^fg^	9.96 ± 0.86 ^fgh^	2.92 ± 0.07 ^def^	23.06 ± 0.46 ^f^
M5	48.85 ± 1.97 ^f^	0.69 ± 0.45 ^ef^	9.72 ± 1.14 ^gh^	3.41 ± 0.43 ^abcde^	24.26 ± 1.50 ^f^
M10	41.59 ± 0.81 ^g^	1.53 ± 0.65 ^de^	11.11 ± 9.26 ^efgh^	4.14 ± 0.41 ^a^	22.25 ± 0.25 ^f^
M15	40.31 ± 1.95 ^gh^	2.83 ± 0.83 ^bcd^	14.13 ± 1.62 ^cde^	3.74 ± 0.10 ^abc^	27.88 ± 0.22 ^e^
M20	36.56 ± 1.97 ^h^	3.12 ± 0.42 ^bc^	14.94 ± 1.25 ^bcd^	3.12 ± 0.45 ^bcdef^	32.62 ± 0.28 ^d^

Values in the same column followed by different superscript letters are significantly different (*p* < 0.05). *L**, lightness; *a**, redness; *b**, yellowness. The wheat flour was substituted by mealworm powder at weight ratios of 0% (M0), 5% (M5), 10% (M10), 15% (M15), and 20% (M20), respectively.

**Table 6 foods-11-04057-t006:** Texture parameters of the bread with various mealworm powder substitution levels.

	Hardness (g)	Resilience	Cohesiveness	Gumminess (g)	Chewiness (g)
GB/T 35869-2018					
M0	462.277 ± 15.660 ^d^	0.525 ± 0.006 ^abc^	0.894 ± 0.003 ^a^	390.893 ± 9.25 ^ef^	393.137 ± 13.640 ^de^
M5	445.152 ± 37.143 ^d^	0.536 ± 0.013 ^abc^	0.871 ± 0.033 ^ab^	455.407 ±42.142 ^d^	465.092 ± 27.649 ^c^
M10	430.077 ± 13.095 ^d^	0.459 ± 0.010 ^def^	0.843 ± 0.017 ^ab^	384.599 ± 0.618 ^ef^	377.371 ± 18.980 ^e^
M15	813.563 ± 6.136 ^b^	0.411 ± 0.006 ^fg^	0.842 ± 0.021 ^ab^	588.502 ± 37.442 ^c^	599.884 ± 21.579 ^b^
M20	743.551 ± 35.999 ^c^	0.356 ± 0.016 ^g^	0.564 ± 0.021 ^c^	419.613 ± 35.926 ^de^	446.603 ± 11.875 ^cd^
GB/T 14611-2008					
M0	415.030 ± 3.271 ^de^	0.542 ± 0.002 ^ab^	0.832 ± 0.048 ^ab^	346.298 ± 21.369 ^fg^	355.844 ± 44.780 ^ef^
M5	350.119 ± 8.804 ^f^	0.508 ± 0.001 ^bcd^	0.868 ± 0.022 ^ab^	303.811 ± 3.051 ^g^	300.932 ± 9.839 ^fg^
M10	354.105 ± 11.235 ^ef^	0.481 ± 0.024 ^cde^	0.862 ± 0.023 ^ab^	305.168 ± 1.287 ^g^	309.441 ± 7.584 ^fg^
M15	1173.597 ± 44.154 ^a^	0.429 ± 0.012 ^ef^	0.801 ± 0.021 ^b^	946.336 ± 7.275 ^a^	921.186 ± 22.580 ^a^
M20	844.396 ± 24.466 ^b^	0.418 ± 0.011 ^f^	0.801 ± 0.028 ^b^	676.114 ± 6.133 ^b^	660.720 ± 27.067 ^b^
Bread maker					
M0	324.915 ± 19.478 ^fg^	0.571 ± 0.011 ^a^	0.889 ± 0.017 ^a^	288.923 ± 19.030 ^gh^	276.196 ± 14.118 ^gh^
M5	262.817 ± 7.039 ^gh^	0.566 ± 0.031 ^a^	0.890 ± 0.028 ^a^	234.035 ± 13.565 ^hi^	228.301 ± 15.000 ^hi^
M10	179.673 ± 4.523 ^ij^	0.529 ± 0.028 ^abc^	0.881 ± 0.019 ^a^	158.232 ± 0.582 ^jk^	151.995 ± 6.261 ^jk^
M15	239.791 ± 3.987 ^hi^	0.488 ± 0.018 ^bcd^	0.885 ± 0.005 ^a^	212.212 ± 4.702 ^ij^	205.552 ± 5.555 ^ij^
M20	141.572 ± 7.248 ^j^	0.499 ± 0.005 ^bcd^	0.868 ± 0.004 ^ab^	122.887 ± 6.080 ^k^	118.243 ± 7.481 ^k^

Values in the same column followed by different superscript letters are significantly different (*p* < 0.05). The flour was substituted by mealworm powder at weight ratios of 0% (M0), 5% (M5), 10% (M10), 15% (M15), and 20% (M20), respectively.

## Data Availability

Data are not available in public datasets, please contact the authors.

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
