# Peer review of "Effect of Mealworm Powder Substitution on the Properties of High-Gluten Wheat Dough and Bread Based on Different Baking Methods"

_foods, 2022, doi:10.3390/foods11244057_

Round 1

Reviewer 1 Report

There is an urgent need for substitute protein sources to replace animal protein production in order to deal with the growth in the global population, the loss of agriculture, and global climate change. Vegetable-derived proteins were chosen as the first possibilities, however, due to their low protein content and availability of other essential amino acids, plants are not the ideal option. Insects have advantages over conventional livestock and plant-based proteins such as being more environmentally friendly, requiring less feed and land, and having a higher food conversion ratio. In this study, high-gluten wheat flour was mixed with dried meal-worm powder at different concentrations (0%, 5%, 10%, 15%, and 20%) to investigate its impact on the pasting, farinograph, extensograph qualities, and microstructure of the dough.

Mealworm utilization in food is a new area, thus there are gaps in the literature. The authors hypothesized that mealworm substitution would affect the pasting characteristics, and farinograph and extensograph properties of the wheat dough, resulting in bread quality change. These results are evident as the gluten network will deteriorate with the addition of mealworm powder. Therefore, there is a problem in addressing the main question, hypothesis and solutions for the gaps. Some studies in the literature investigate the effect of mealworm flour addition on the nutritional properties of bread. In the paper that the authors also used as a reference (Roncolini et al., 2019), the authors investigated the effect of mealworm powder incorporation on bread doughs. In that research, the researchers performed some rheological analysis, including farinograph, alveograph and amylograph. In the submitted manuscript, the authors did not give an improvement to the literature.

There are still some papers that should be put on the reference list.

Gaglio et al., 2021 https://doi.org/10.1016/j.ifset.2021.102755

Cozmuta et al., 2022; https://doi.org/10.1016/j.jff.2022.105310

Author Response

Response: Thanks for the comments. In the process of designing the experiment and preparation of the manuscript, we also usually ask ourselves about the novelty and improvement point compared to the previous studies. We still found some points that needs to be clarified in our study, which is progress compared to previous studies.

  • That the gluten network was deteriorated by mealworm powder, while the specific volume of the bread was significantly increased. Even though the pasting and farinograph properties were decreased gradually, owing to the starch and gluten dilution effect, the elasticity and flexibility of the high-gluten wheat dough were increased.
  • There is some reverse conclusion about the effect of mealworm powder on the bread quality. We intended to confirm the trend by three different baking method. Roncolini et al. observed a similar increase in the specific volume after substituting soft wheat flour with 5% and 10% mealworm powder; they found that the 5% substitution level achieved the highest specific volume, but they did not study a substitution level higher than 10%. It is known that mealworm powder is rich in not only protein but also lipids, which can be adsorbed at the gas cell–dough interface and then increase the gas retention during leavening. However, some previous studies have shown an opposite effect of mealworm powder on the bread specific volume. Khuenpet et al. found that the specific volume of bread decreased gradually with an increase in meal-worm substitution up to 15% and that the hardness of bread increased by approximately four times compared with the control bread. González et al. used mealworm powder to replace 5% wheat flour and observed a significantly decreased specific volume com-pared to the control.

In our study, we confirmed that the bread quality was improved at 5% and 10% mealworm substitution level and high substitution level would deteriorate the bread quality.

  • In previous studies, the effect of mealworm powder on the bread quality has varied, which might be ascribed to the various production methods and manual factors during bread making. An automatic bread maker is a common bread-making appliance, which can be applied to bread making in the laboratory to eliminate any manual factors. We found that the automatic bread maker method might be an effective method for bakery studies to avoid manual variance. The bread quality change showed similar trend among the three different baking methods.

The suggested literatures have been added properly. 

Reviewer 2 Report

This article "Effect of mealworm powder substitution on the property of 2 high-gluten wheat dough and bread by different making methods” was revised and has a novelty. A similar work “Effect of Partial Substitution of Flour with Mealworm (Tenebrio molitor L.) Powder on Dough and Biscuit Properties” has been published in the Foods. My comments see below:

Title: “by different making methods”: this part is not clear.

Abstract:

·       Abstract needs to add more numerical data.

·       Lines 10-11: What is the reason for choosing these two temperatures and why the same times were not used for the two temperatures of 120 and 130 degrees Celsius?

·       The abstract is very concise. Please explain more and more about the scientific study results.

·       It is better to express the statistical comparison between the 6 treatments.

Keywords: Please add “baking methods” or “type of bread”.

Abbreviation:

·       Please provide “Abbreviation section consequent the Keywords

Introduction:

It is good.

Why has high-gluten flour been chosen?

Materials and methods: d

Line 118: the thickness of bread? It was a flatbread?

Line 137, 147: Please provide an appropriate reference: 10.1177/10820132211063972

·              Line 148: The compression test was explained. But the TPA test parameter has been presented. Have two cycles of compression been done?

           Line 155: Which method was used to determine significant 158 differences between mean values? Duncan, LSD…?    

Results:

Line 279: why does the specific volume increase at a low level of mealworm?

The discussion about bread-making methods results was deep. Please add more discussion.

Line 304: machinefor>>machine for

Texture analysis: this part needs more discussion and comparison with the previous investigations. 

Conclusions:

·       Conclusion is not perfect. Please express the detail of your study as a comparing approach.

References: It is OK.

 The English was good and need some minor revision.

Author Response

This article "Effect of mealworm powder substitution on the property of 2 high-gluten wheat dough and bread by different making methods” was revised and has a novelty. A similar work “Effect of Partial Substitution of Flour with Mealworm (Tenebrio molitor L.) Powder on Dough and Biscuit Properties” has been published in the Foods. My comments see below:

Title: “by different making methods”: this part is not clear.

Response: Thanks for your suggestion. We have changed the title to “Effect of mealworm powder substitution on the properties of high-gluten wheat dough and bread based on different baking methods”.

Abstract:

Abstract needs to add more numerical data.

Response: We have revised the abstract and added the more numerical data in abstract.

Lines 10-11: What is the reason for choosing these two temperatures and why the same times were not used for the two temperatures of 120 and 130 degrees Celsius?

Response: Thank you for your question. We didn’t find the description of the temperature in lines 10-11. We have revised the abstract carefully.

The abstract is very concise. Please explain more and more about the scientific study results.

It is better to express the statistical comparison between the 6 treatments.

Response: Thank you for your suggestions, we have added statistical comparisons.

Keywords: Please add “baking methods” or “type of bread”.

Response: Thank you for your suggestion, we have added “baking methods” to the keywords.

Abbreviation:

Please provide “Abbreviation section consequent the Keywords

Response: All abbreviations has been defined at first mention, and those in figure and table legends as figures and tables are explained to stand alone from the text.

Introduction:

It is good.

Why has high-gluten flour been chosen?

Response: We choose high gluten flour for bread making because it has high protein content and can easily form gluten network, which can increase the air retention of the dough after fermentation and good for the quality of bread. Bread is commonly made by high-gluten wheat flour. The whole project of the master thesis of the first author is to study the effect of mealworm powder on the low, middle, and high-gluten wheat flour and the representative baking products. Bread was chosen as the representative baking product of high-gluten wheat flour.

Materials and methods:

Line 118: the thickness of bread? It was a flatbread?

Response: Thank you for your question, as shown in Figure 5, we make the bread into a hemisphere with a thickness of about 5 cm by GB/T 35869—2018 Rapid-baking test method and GB/T 14611–2008 Straight dough method. The bread by automatic bread maker is cubic with a thickness of about 10 cm.

Line 137, 147: Please provide an appropriate reference: 10.1177/10820132211063972

Response: We refer to this literature and make reasonable references in the article.

Line 148: The compression test was explained. But the TPA test parameter has been presented. Have two cycles of compression been done?

Response: We tested the bread by TPA model with two cycles of compression. We have revised the description in Line 159-161.

Line 155: Which method was used to determine significant 158 differences between mean values? Duncan, LSD…?   

Response: We added the method for determining significant differences between mean values in Line 166-167. “The Tukey’s test at the P < 0.05 was used to determine significant differences between mean values.”

Results:

Line 279: why does the specific volume increase at a low level of mealworm?

Response: Even though a subsequent decrease in the pasting parameters was observed due to starch dilution. The farinograph properties corresponded to a weak gluten network formed through the dilution of gluten by the replacement of wheat flour with non-gluten ingredients. The high-gluten dough showed increased stretching resistance and extensibility. It can be inferred that mealworm powder can act as a “plasticizer” in the gluten network that might contribute to the decreased strength and increased elasticity and flexibility of the dough network, resulting in increased gas retention ability at 5% and 10% mealworm substitution levels. We have added the discussion in Line 294-301.

The discussion about bread-making methods results was deep. Please add more discussion.

Response: We have added discussions to compare the result differences among the different baking methods in Line 349-355.

Line 304: machinefor>>machine for

Response: We have revised it.

Texture analysis: this part needs more discussion and comparison with the previous investigations.

Response: We have added discussion with previous investigation in Line 357-363.

Conclusions: Conclusion is not perfect. Please express the detail of your study as a comparing approach.

Response: We have revised our conclusion accordingly.

References: It is OK.

Response: We have carefully checked again after adding some references.

The English was good and need some minor revision.

Response: We tried our best to improve the manuscript. The paper has been carefully revised by a professional language editing service to improve the grammar and readability. 

Reviewer 3 Report

The authors present a comprehensive evaluation of the quality of bread prepared by replacing part of the flour with mealworm powder. The dough properties (development time, maximum consistency, stability, extensibility and resistance to deformation) and the bread sensory attributes were evaluated with various appropriate techniques. The possible dependence on bread making method was also tested.

Overall, the manuscript is well focused on its aim. The sections of the manuscript are clear and well written, and results are discussed with reference to the relevant literature. 

The manuscript is not only of food technological interest, but also has the merit of showing, using a chemical-physical approach, that mealworm powder exerts a plasticizing effect on the gluten network.

Author Response

Response: Thank reviewers for taking time out of their busy schedule to review my manuscript. We have revised the manuscript according to other reviewers’ suggestions and carefully checked our manuscript to improve it.

Round 2

Reviewer 2 Report

The revised manuscript is acceptable.